# Situating trade-offs: Stakeholder perspectives on overtreatment versus missed diagnosis in transition to Xpert MTB/RIF Ultra in Kenya and Swaziland

**Muthoni Mwaura[1], Kekeletso Kao[2], Jesse Wambugu[2], Andre Trollip[2], Welile Sikhondze[3], Eunice Omesa[4], Sindi Dlamini[5], Nompumelelo Mzizi[3], Muyalo Dlamini[3], Busizwe Sibandze[3], Brian Dlamini[3], Heidi Albert[2], Wybo Dondorp[1], Nora Engel[1]***

1 Department of Health, Ethics & Society; Care and Public Health Research Institute (CAPHRI), Maastricht University, Maastricht, The Netherlands, 2 Foundation for Innovative New Diagnostics (FIIND), Geneva, Switzerland, 3 National TB Control Program, Ministry of Health, Mbabane, Swaziland, 4 National Tuberculosis, Leprosy, and Lung Disease Program, Ministry of Health, Nairobi, Kenya, 5 Swaziland Health Laboratory Service, Ministry of Health, Mbabane, Swaziland

* n.engel@maastrichtuniversity.nl

**Data Availability Statement:** There are ethical restrictions of sharing a de-identified data set publicly. Our interview data contain potentially

## Abstract

Implementing new diagnostics in public health programs can involve difficult trade-off decisions between individual patient benefits and public health considerations. Such decision-making processes are often not documented and may not include engagement of affected communities. This paper examines the perspectives of stakeholders on the trade-off between over-treatment and missed diagnosis captured during decision-making workshops on the transition from use of Xpert MTB/RIF to diagnose tuberculosis to Xpert MTB/RIF Ultra in Kenya and Swaziland. Xpert MTB/RIF Ultra has an overall increase in sensitivity but a decrease in specificity when compared to its predecessor. We conducted a qualitative study using four focus group discussions with a total of 47 participants and non-participant observation. The analysis reveals how participants deemed Xpert MTB/RIF Ultra's reduced specificity vis-à-vis its increased sensitivity to be an acceptable trade-off. The way participants assessed this trade-off was shaped by their experiences with the general uncertainty of all diagnostic tests, alternative testing options, historical evolution of diagnostic practices, epidemiological factors and resource constraints. In assessing the trade-off community and individual benefit and harm was frequently discussed together. Qualitative research on stakeholder engagement activities for diagnostic development and implementation can identify everyday experiences and situate assessments and perspectives of key stakeholders and as such aid in decision-making, improving implementation as well as patient outcomes. Further research is needed on the intended and unintended consequences of such engagement activities, how findings are being incorporated by decision-makers, and the impact on programmatic implementation.

identifying or sensitive information. These restrictions were imposed by the ethics committee. Public deposition of the data would compromise interviewee privacy which we promised during the informed consent procedure. Participants did not consent to having their data be made publicly available. Data are available upon individual request with the necessary editing to preserve anonymity of the study participants. Contact the ethics review board at fhml-rec@maastrichtuniversity.nl.

**Funding:** This project is supported by the Department for International Development, UK government. The URL of the funder's website is www.gov.uk/government/organisations/department-for-international-development. Grant number is DFICORE-03, and the funding was received by the Foundation for Innovative New Diagnostics. The funders had no role in study design, data collection and analysis, decision to publish, or preparation of the manuscript.

**Competing interests:** I have read the journal's policy and the authors of this manuscript have the following competing interests: FIND collaborated with Cepheid and Rutgers to develop the Xpert MTB/RIF Ultra cartridge. FIND also led the evaluation studies to get the cartridge endorsed by WHO thus increasing acceptability of the cartridges in the countries of intended use. This does not alter our adherence to PLOS ONE policies on sharing data and materials.

## Introduction

New diagnostics are an important part of the global response to tuberculosis (TB). Decisions to implement new diagnostics as part of public TB programs often involve difficult trade-off decisions between benefit and harm to both the patient and community based on test performance, characteristics, and public health and implementation concerns. Yet, such decision-making processes are often not documented and may not include engagement of affected communities. This paper examines the perspectives of stakeholders on the trade-off between overtreatment and missed diagnosis captured during decision-making workshops on the transition from use of Xpert MTB/RIF to Xpert MTB/RIF Ultra in Kenya and Swaziland.

At the time of its launch in 2010, WHO recommended that the Xpert MTB/RIF (Cepheid Inc., Sunnyvale, CA) be used as the initial diagnostic test for pulmonary TB in adults (and later it was recommended for use in children and in diagnosing certain forms of extrapulmonary TB) [1]. The test detects Mycobacterium tuberculosis (MTB) and resistance to anti-TB drug rifampicin (RIF) with an automated, cartridge-based platform with a two hours machine turn-around time [2]. Although the test was a monumental step forward for TB diagnostics, its sensitivity among people living with HIV (PLHIV), extrapulmonary TB (EPTB) and children is limited due to the low concentration of mycobacteria in the clinical samples of these individuals [3][4][5]. To address this, Cepheid developed a new assay called the Xpert MTB/RIF Ultra which can diagnose TB in samples with a lower concentration of bacteria [2][6]. This increase in sensitivity has been accompanied by a decrease in specificity, potentially presenting a trade-off between overtreatment and missed diagnoses.

Although handling trade-offs is a common feature of diagnostic development and public health practice, little research has been done on how different TB stakeholders view such diagnostic trade-offs and how countries make decisions about these [7]. For global health [8], stakeholder engagement is considered an important tool in improving decision making on policy and guideline development for patient-centred care [9]. Communities can be defined by geographical proximity, shared identity, professional group or participants of an engagement event. Their engagement can range from merely taking consent, to educational activities or explicitly democratizing knowledge production processes [8]. Yet, detailed methods, frameworks and outcomes of such engagements for policy development are rarely published [10].

In 2017, the national tuberculosis programs of Kenya and Swaziland, with support from the Foundation for Innovative New Diagnostics (FIND), hosted workshops with country stakeholders in order to review and contextualize the above recommendations on Xpert MTB/RIF Ultra by WHO, and if necessary, adjust the national TB algorithms. This paper is based on qualitative research conducted during these workshops. The aim of the research was to examine the views and norms of multiple TB stakeholders on the trade-off between overtreatment versus under diagnosis of TB, and to understand the role qualitative research can play in engaging in-country stakeholders during the launch and roll-out of new TB diagnostics.

### The trade off between overtreatment and missed diagnosis

The accuracy of diagnostic tests is evaluated as sensitivity (the potential of a diagnostic to provide a positive result in people who really have the disease) and specificity (how often the diagnostic provides a negative result in people who do not have the disease. When compared to Xpert MTB/RIF, the overall sensitivity of Xpert MTB/RIF Ultra is higher by 5% [6]. Specifically, increases in sensitivity were seen in smear-negative patients (+17%), PLHIV (+12%), patients with TB meningitis (+50%) and children (+24%) [6][11].

The main contributor to the increase in Xpert MTB/RIF Ultra's sensitivity is the new category of results, 'trace'. Trace results refer to the lowest bacillary burden that is detectable in a

sample by Xpert MTB/RIF Ultra. Yet, due to the low levels of TB bacilli that are being detected no definite answer can be given on whether the patient is also rifampicin resistant [6]. The new patient category of Xpert MTB/RIF Ultra is therefore 'MTB detected, trace, RIF indeterminate' [12][13].

The increase in sensitivity provided by the trace results adds to the decrease in the specificity of Xpert MTB/RIF Ultra. When contribution of trace results is factored into the Xpert MTB/RIF Ultra results, specificity as compared to Xpert MTB/RIF, decreased by 3.2%, with the lowest dips among patients with a history of TB in countries with high TB incidence (-8%) [6]. In a modelling exercise comparing the impact of Xpert MTB/RIF to Xpert MTB/RIF Ultra it was found that although Xpert MTB/RIF Ultra may be able to detect 1 additional TB case per 100 to 1000 individuals and prevent 1 additional TB death per 500 to 10,000 individuals, it may also falsely detect 1 TB case per 40 to 70 individuals, and lead to 10 to 500 unnecessary treatments for each TB death prevented [14].

The WHO [13] recommends that the interpretation of Xpert MTB/RIF Ultra results should be the same as for the predecessor Xpert MTB/RIF results [15], with exception for the interpretation of 'trace' results. Here, WHO [13] recommends: (1) positive 'trace' results should be considered a true positive result among PLHIV, children and extra-pulmonary specimens; (2) positive 'trace' results among individuals not at risk of HIV should be retested with fresh specimen; (3) among patients without a recent history of TB, a second 'trace' result can be diagnosed as pulmonary TB; and (4) individuals that test positive for 'trace' need further confirmatory tests to determine rifampicin resistance. The concessional price for Xpert MTB/RIF Ultra cartridge will remain the same for eligible countries—at USD 9.98—and existing Xpert MTB/RIF equipment can be used to run the new cartridge [15]. As such, in deciding to adopt Xpert MTB/RIF Ultra, countries need to evaluate what is an acceptable number of unnecessary treatments for every TB patient correctly diagnosed and treated within their contexts.

Based on the above recommendations from WHO, the Global Laboratory Initiative (GLI)– a Working Group of the Stop TB Partnership–developed a guide to inform the transition from the use of Xpert MTB/RIF to Xpert MTB/RIF Ultra cartridges. The guide recommends the development of an algorithm that specifies how to interpret and confirm results of patients with a recent history of TB as well as the need to ensure that mechanisms are established to support repeat confirmatory testing [2]. Considerations of local TB prevalence, existing empiric treatment practices for those patients who test negative on Xpert MTB/RIF and the impact on patients of delayed diagnosis of rifampicin resistance due trace categories should also feed into the decision-making [2][14][16].

Clinical epidemiologists in the Philippines have noted that when deciding to adopt a new public health screening test that presents a trade-off between benefit and harm, five criteria should be satisfied: "(1) the burden of illness should be high; (2) the test for screening and confirmation should be accurate; (3) early treatment (or prevention) must be more effective than late treatment; (4) the test(s) and treatment(s) must be safe meaning that the balance of benefits and harms for those tested must be positive; and (5) the material cost of the screening strategy must be commensurate with the potential benefit"[17]. Similarly, in the debate between specificity and sensitivity related to screenings, ethicists have noted that both false positives and false negatives are problematic [18]. While false negatives can create a false sense of security and consequently delay treatment for the individual, false positives can create unnecessary anxiety and stigma, and lead to unnecessary treatment. On a societal level, both false positives and false negatives can lead to a decrease in the public's trust in the health care system [18]. In determining if false positives or negatives are more acceptable in screening programs, ethicists suggest considering several factors: (1) the disease being targeted; (2) the consequences for the

individual and the society; and (3) the access and efficacy of available treatments [18]. Both these set of criteria were developed for public health screening practices and not diagnostics. As such, stakeholder perspectives on how to balance benefits and harms of diagnostic trade-offs might differ based on the disease and country context.

## Materials and methods

### Setting

This study was positioned within two large stakeholder meetings in Kenya and Swaziland hosted by the National Tuberculosis, Leprosy and Lung Disease (NTLD) program and the National Tuberculosis Control Program (NTCP) respectively, facilitated by technical support from FIND. Both Kenya and Swaziland have high rates of TB and HIV coinfection and MDR-TB [1]. Both countries were early adopters of Xpert MTB/RIF [19][20] as the initial diagnostic test for all presumptive TB patients, based on the WHO recommended algorithm shown in Fig 1 [2]. Despite having similar diagnostic algorithms, access to Xpert MTB/RIF testing varies quite significantly with 26% and 82% of presumptive TB patients being tested by Xpert MTB/RIF in Kenya and Swaziland respectively [1]. At the time of the workshop, Kenya had already decided to adopt Xpert MTB/RIF Ultra in the place of Xpert MTB/RIF [13], and the purpose of the workshop was to develop a transition plan, while in Swaziland, a decision to adopt Xpert MTB/RIF Ultra was to be made following the workshop. The meeting in Kenya took place from the 31st of July to the 1st of August 2017, and the meeting in Swaziland from the 19th to the 21st of September 2017.

The two-day workshop in Kenya included an overview of Xpert MTB/RIF Ultra, the current national TB diagnostic algorithm, and the development of a transitioning plan, tools and indicators for the new test. The FGDs for this study took place on the second of the workshop. A similar two-day agenda was followed in Swaziland, with participants reviewing the national TB program and diagnostic algorithm, research on Xpert MTB/RIF Ultra, and modelling exercises on Xpert MTB/RIF Ultra in various Swazi settings. The FGD in Swaziland took place at the end of the workshop's second day.

### Participants

The Kenyan NTLD program identified and invited 20 stakeholders to attend the workshop, including clinicians, laboratory technicians, national and local TB program coordinators, TB patient advocates from local organizations, and non-governmental organisation representatives. Three FGDs (N = 17) were conducted, each with 5–6 participants purposefully distributed to ensure adequate professional representation. Each discussion lasted about an hour each. In Swaziland, all 30 stakeholders that were invited by the NTCP participated in the large FGD. The aforementioned professions were represented in this FGD, which lasted about an hour.

### Data collection

The study methodology is rooted in the interpretivist perspective [21]. Through this lens, individuals attach subjective meanings to objectively verifiable facts [21]. Identifying and analyzing these meanings offers rich and nuanced data that can inform programmatic decision-making at all levels. The present study used FGDs as the main method of data collection. FGDs are an accepted and widely-used method for qualitative research that allow for the in-depth exploration of participant views and understandings. A set of predetermined questions for the FGDs was designed prior to the meetings (see questions in S1 Appendix). The main topics included

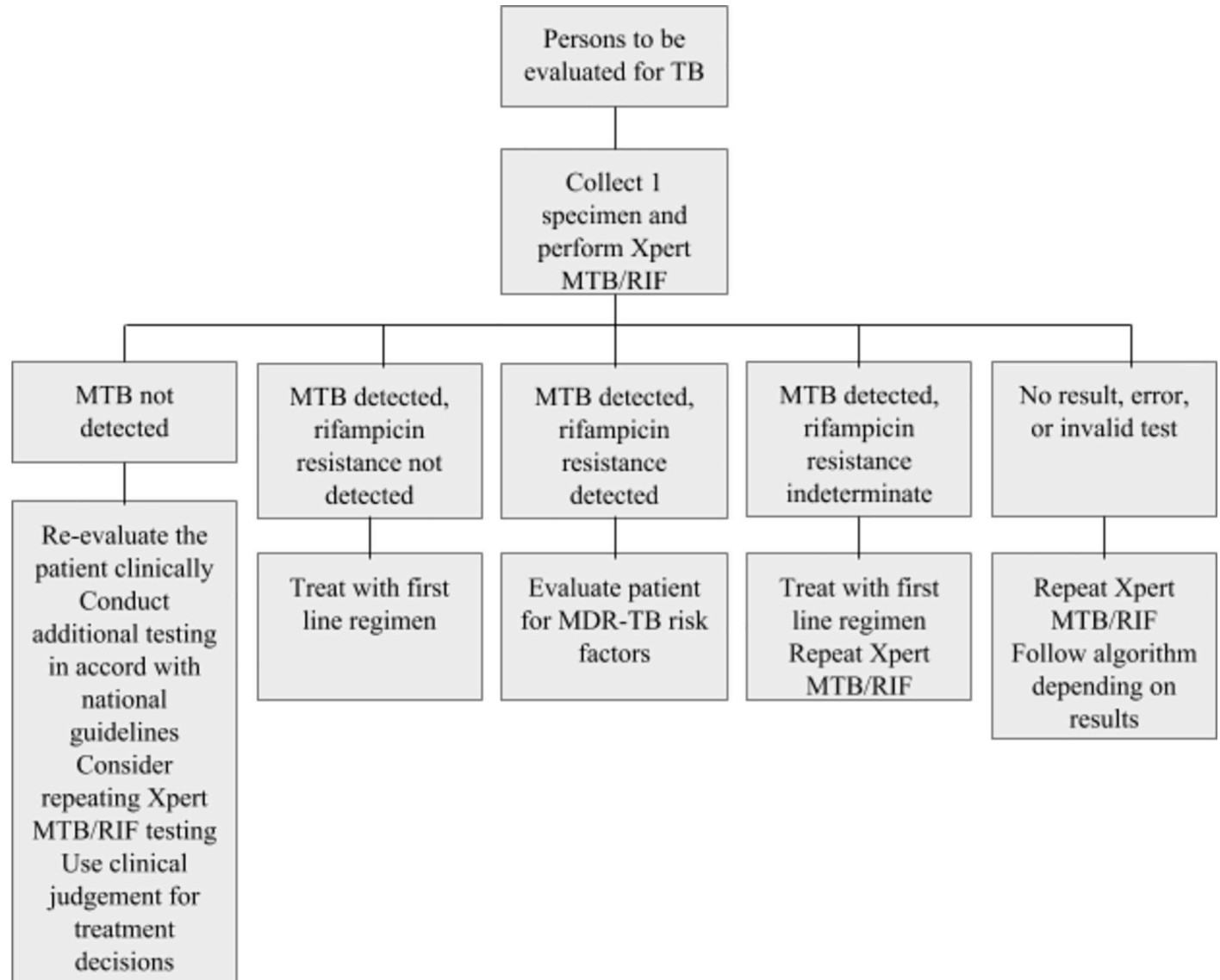

**Fig 1. WHO recommended algorithm for diagnosing all presumptive TB patients with Xpert MTB/RIF.**

challenges and successes of diagnosing TB, implementing new diagnostic tools, and the trade-off between sensitivity and specificity of Xpert MTB/RIF Ultra. By keeping the questions open-ended, participants were encouraged to share their subjective experiences, opinions, ideas and feelings regarding these topics. The first author, a social scientist trained in qualitative methodology, moderated the discussions.

In Kenya, each FGD began with an exercise that prompted participants to note down on a piece of paper any challenges and successes that they have experienced in TB diagnosis within their settings. In Swaziland, in lieu of noting down on a piece of paper, participants were requested to verbally share challenges that they have experienced in TB diagnosis, while the moderator noted down responses on a flip chart at the front of the room. This exercise was modified to accommodate the larger size of the group as well as the limited time available to

carry out the discussion. In both settings, this exercise provided a window through which participants were able to contextualize and discuss their realities surrounding TB diagnosis. The remaining discussion was devoted to unpacking as many points as possible in greater depth as a group. Where possible, participants were prompted to consider how Xpert MTB/RIF Ultra would impact that particular challenge or success. The moderator did not explicitly frame any of the discussions in terms of the tradeoff but allowed the topic to come up naturally as each discussion progressed. This was done intentionally in order to gauge whether a tradeoff between overtreatment and underdiagnoses was even perceived to exist by the participants, and if so, how it was interpreted in the context of their day-to-day realities.

### Data analysis

With the permission of the participants of the Kenyan cohort, the FGDs were electronically recorded. The recordings were transcribed, with all identifying information redacted for confidentiality purposes. For the Swaziland arm of the research, the FGD was not recorded, but descriptive notes were taken and later member-checked with the participants following the discussion. By tailoring the FGD discussion in Swaziland to accommodate a larger number of participants, the ability to spend extensive time unpacking a wider range of issues was lost. As such, the data obtained from this study site was not as rich as that of the Kenyan site. An inductive analytical process was used to identify themes and patterns emerging from both sets of data related to the interpretation of the trade-off.

### Ethics

Review by an institutional ethics committee was not sought since the FGDs represented voluntary discussions within a routine workshop and were approved as such by the National TB Programs. For the Kenyan FGDs, an informed consent form was designed using the Kenyan Institutional and Ethics Committee (IREC) guidelines. Prior to each discussion, the participants were given an opportunity to read through the form, ask the FDG moderator questions, and sign it. The FGD in Swaziland served as a mid-meeting evaluation. As such, national research protocol did not require the use of an informed consent form in this particular setting, and for this reason, the discussion was not recorded. Participants were informed about the purpose of the discussion and their rights beforehand and the discussion was structured along the same guiding questions as the other FGDs. Employees of FIND were not present during the FGDs in Kenya. In Swaziland, FIND employees were present during the FGD but as note-takers and not discussion moderators.

### Results

In each of the FGDs, the topic of the trade-off between sensitivity and specificity came up naturally and was further probed by the moderator. It was common for participants to assess and rationalize the trade-off within the context of the challenges they faced. In both countries, participants deemed Xpert MTB/RIF Ultra's reduced specificity vis-à-vis its increased sensitivity to be an acceptable trade-off. The results below highlight how the specific way of assessing the trade-off related to Xpert MTB/RIF Ultra was embedded within the participants' understandings of and experiences with the general uncertainty of all diagnostic tests, alternative testing options and historical evaluation of diagnostic practices, programmatic gaps, the safety of the test, and unintended consequences. Below, these findings are explored in further detail.

## General uncertainty: No TB test is perfect

The most common rationalization used was that no TB test is perfect and that before Xpert MTB/RIF, they may have already been over-treating individuals. A clinician from a high TB/HIV burden county in Kenya stated,

"*For me, I think this question of wrongfully treating people has to be looked at in a bigger picture both currently and historically . . .we have come a long way in terms of TB diagnostics, and clinical decision-making, based on available technology . . .So if we look at the historical trend of the unnecessarily treated, I would expect that this is generally going down. And probably we have the qualms because we are merely comparing Ultra to Gene Xpert.*"

Similarly, speaking to how current laboratory tests are all based on some level on uncertainty, a lab technician from Kenya's National Tuberculosis Reference Laboratory noted,

"*In the lab with each test we have a measure of uncertainty . . .so to me, it is just as the normal routine like all other tests that we do. So having [the sensitivity] being so nice, we go for it.*"

## Alternative testing options and existing diagnostic practices

As mentioned above, Xpert MTB/RIF has imperfect sensitivity among PLHIV and obtaining a sufficient volume of good quality sputum from EPTB and children to run a Xpert MTB/RIF test has been a challenge [11]. Indeed, these challenges were regularly discussed by study participants, many of whom came from settings with high rates of TB and HIV coinfection (31% and 70% of TB patients are co-infected with HIV in Kenya and Swaziland respectively) [1], EPTB, and pediatric TB. Improving TB case detection among these hard-to-diagnose patients seemed an important advantage of Xpert MTB/RIF Ultra to these participants. A TB coordinator from a high HIV burden county in Kenya succinctly put it, "*yeah we accept it because we know most of our people will benefit from it.*"

Historically, diagnosis of TB among PLHIV, EPTB and children has predominantly relied upon empirical diagnosis. Although current literature is not clear on this, participants of the study claimed that this reliance on empirical diagnosis decreased with the introduction of Xpert MTB/RIF. Yet, the suboptimal sensitivity of Xpert MTB/RIF further hampered case detection within these patient groups, because clinicians would rely on the (imperfect) molecular method at expense of their empirical acumen. As a TB program county coordinator from Kenya explains,

"*. . .when Xpert came on board, we, the communication initially was, we have gotten a test that is better than smear, that we will have more patients being diagnosed confirmed for TB. And uh what the clinician interpreted was that if you are Xpert negative, then you do not have TB. And I think we lost a huge number of patients then. Because, if we look at our data from about 2, 3 years ago when we started Xpert, the bacteriologically confirmed cases started going up, the [empirically-diagnosed] cases started going down*".

When asked how Xpert MTB/RIF Ultra would address this, some anticipated that empirical diagnosis would further decrease. Others were optimistic that through 'trace' results, the number of bacteriological confirmed cases would increase among these hard-to-diagnose patient groups, because the new assay is capable of detecting these. As a clinician from Kenya notes,

"*There are people we are missing that are falling in that group of clinically-diagnosed—the HIV positive, the children, the people with no history—that can now be picked by trace calls, that we couldn't pick with Xpert. So they would definitely go home as smear negative, Xpert negative. So I am hoping, though the clinical diagnosis will go down, we will still capture a number as bacteriologically confirmed.*"

Here, the limitations of existing testing options, the historical evolution of clinical practices due to new testing options, and the epidemiological factors of certain population groups interact in the assessment of the trade-off related to Xpert MTB/RIF Ultra.

## Experiences with programmatic challenges and scarcity of equipment

The acceptability of Xpert MTB/RIF Ultra was also discussed in light of experiences with Xpert MTB/RIF cartridge stock outs and erratic power supply. In Kenya, stock-outs were said to occur due to a mismatch between the number of cartridges used at the various Xpert MTB/RIF sites and the number of cartridges dispatched from the national program. Participants claimed this mismatch was due to inadequate reporting at local testing centres, ineffective usage of the electronic information management system, and the centralised cartridge supply management system. In Swaziland, stock outs of the Xpert MTB/RIF cartridges were said to occur due to gaps in the supply chain causing delays between the importation and distribution of cartridges and resulting in expired cartridges within the facilities. In both settings, these stock-outs meant a reversion to using sputum microscopy as the main TB diagnostic tool. Since sputum microscopy has a longer turnaround time, patient advocates argued that this regression increases costs to the patient (loss of wages and transportation from having to make additional follow-up visits to the facility) and increases the risk of TB transmission within communities due to diagnostic delays. Similarly, participants in both countries noted erratic power supply as a major challenge in TB diagnosis causing technical errors, low Gene Xpert utilization rates, and patient attrition during diagnosis (because a new sample may be needed to run a fresh test but a patient may be unwilling to return to the facility to provide one).

The Kenya participants saw the transition to Xpert MTB/RIF Ultra as an opportunity to also improve back-up power supply to GeneXpert sites and with it utilization and patient retention rates. When asked how the introduction of Xpert MTB/RIF Ultra could impact stock outs in Kenya, one participant noted with optimism that the introduction of a new technology usually comes with a big supply:,

"*..if the Ultra cartridge [comes] to the ground, probably they would be able to have a wide scope on how they are going to implement, in terms of the rolling out of these cartridges to the facilities to make sure that they are really there. Because definitely, as usual, when any product comes, its comes in a big supply...*"

With regards to erratic power supply, one participant in Swaziland was hopeful that the reduced running time of Xpert MTB/RIF Ultra may yield less errors leading to the completion of more tests in-between power outages. Although other participants believed that these were infrastructural issues that could not be mended with the introduction of a new diagnostic test, this highlights the nuanced ways in which experiences with programmatic challenges, such as availability of testing equipment and infrastructural support systems, play into assessments of diagnostic trade-offs.

## Do no harm

In reasoning whether Xpert MTB/RIF Ultra's trade-off is acceptable, participants discussed whether the reduced specificity of Xpert MTB/RIF Ultra would cause patients any harm. Here, harm was conceptualized in terms of underdiagnosing and overtreating. In Kenya the participants appeared not to be too concerned with over diagnosis. Since they were already treating a large number of patients with no laboratory confirmation of a TB diagnosis, over diagnosing TB with Xpert MTB/RIF Ultra would be no worse than the current practice. In both Kenya and Swaziland, there was a general attitude that over-treating is better than underdiagnosing. Reasons for this included how the latter not only endangered individuals, but also increased risk of transmission within communities. Although the literature suggests to distinguish between harm to/benefits for an individual and harm to/benefits for the community when carrying out an ethical evaluation of a new test [18], this distinction was often not clearly made by participants of this study and the two were often discussed together. For example, in discussing the issue with false negatives, a participant in Kenya notes how individual and community benefits hang together:

> "*But I think the benefits outweigh the disadvantages. . .Thats the way I would look at it, because we have you know 30 more people that we would have missed, who would be dead and you know, transmit TB to other people.*"

Similarly, in discussing issues with long turnaround times, a patient advocate stated,

> "*So you are delaying diagnosis, you are losing the patient, they are spreading TB, we are really just perpetuating the transmission of tuberculosis, as well as, you know not treating the patient.*"

Although false positives were deemed the lesser evil by most participants, over-treating was not perceived to be without faults. The most frequent consideration was the unknown effect of overtreatment in regards to the development of drug resistance. Over-treating also did not sit well with some due to the side effects of some anti-TB medication, the costs incurred by patients to obtain the medication [22][23] and the long-term effects of being diagnosed with a stigmatized disease. Others, however, considered how over-treatment may be beneficial in case of latent TB, with one clinician in Kenya noting,

> "*. . .this is not anti-cancer medication which is going to in itself be able to kill you . . .it has its own benefits [as] a couple of the patients who will unnecessarily be on treatment could also have TB infection rather than TB disease and therefore could still benefit from that . . .so from a public health perspective, I do not think I am too worried. . .*"

Here, the discussion on how much harm is caused to individuals and communities shaped the debate between over-treating patients due to the reduced specificity of Xpert MTB/RIF Ultra and underdiagnosing TB cases due to the imperfect sensitivity of Xpert MTB/RIF. In Kenya a few program coordinators and clinicians considered ways in which use of the Xpert MTB/RIF cartridge can be maintained while using the Xpert MTB/RIF Ultra cartridge only for those patient groups within which sensitivity was increased i.e. PLHIV, EPTB and children thus potentially reducing the possibility of misdiagnosis. This idea was soon discarded due to the potential confusion that would arise with having two similar cartridges within the diagnostic algorithm.

Among ethicists, the acceptable level of harm to individuals caused by a test depends on the context in which the test is being used [18]. In general, specificity is of more importance if the test is being used for screening purposes among a yet unburdened population. Sensitivity is of greater value if the test is being used as a final diagnostic tool. From a normative perspective, false positives within a diagnostic and treatment care setting are not as problematic as false negatives in this same setting. Although participants of this study did not emphasize the difference between screening and diagnostic settings in their assessments of the trade-off, the fact that most participants worked in clinical care and diagnostic settings may help explain why there was a general attitude that false positives are less problematic than false negatives.

## Domino-effect: Thinking through unintended consequences

In Swaziland, some participants expressed other concerns with the reduced specificity of Xpert MTB/RIF Ultra. These included the risk of litigation from patients for a false diagnosis and reduced trust in the healthcare system. Illustrating the latter, a participant expressed that through word of mouth, an individual's false diagnosis could diminish the trust those within his/her network have in future diagnoses. This could have unintended effects on adherence to medication for various diseases or increase attrition of presumptive TB cases.

## Discussion

This paper examines the perspectives of stakeholders on the trade-off between over-treatment and missed diagnosis of TB and rifampicin resistance during the transition from use of an existing diagnostic test, Xpert MTB/RIF, to a new one, Xpert MTB/RIF Ultra. During decision-making workshops in Kenya and Swaziland, we studied how TB stakeholders from different contexts and professions interpret a uniquely sensitive issue: is overtreatment better or worse than under-treatment; is reduced specificity an acceptable cost of increased sensitivity? In both countries, participants deemed Xpert MTB/RIF Ultra's reduced specificity vis-à-vis its increased sensitivity to be an acceptable trade-off.

This paper brought to light the nuanced ways in which participants evaluated this particular trade-off, demonstrating the value of qualitative methodology in TB diagnostics research [7]. These discussions showed that the assessment of trade-offs is shaped by the everyday experiences with the general uncertainty of earlier diagnostic tests, the availability of alternative testing options (including how expensive and timely they are), the historical evolution of clinical practices (in this case existing concerns of overtreatment and misdiagnosis are fueled by dominance of empirical treatment and concerns of losing clinical acumen when increasingly relying on suboptimal test for hard to diagnose patient groups), and epidemiological factors of particular patient groups that are difficult to diagnose using existing options; but also by programmatic gaps and resource constraints (in this case cartridge stock outs and erratic power supplies). These aspects interact in the considerations around the trade-off and the unintended consequences. Contrary to ethics literature where harm/benefit of a test for the individual and for the community are separately assessed [18], the results reveal how harm/benefit for individual and communities hang together and are being discussed as such.

A recent analysis of the roll-out of Xpert MTB/RIF suggested that a comprehensive implementation approach needs to be adopted in order to maximize expected benefits of new technologies [24]. A key recommendation from this research was the early engagement of in-country stakeholders in product launch and roll-out [24]. Such stakeholder engagement for diagnostic development and implementation must go beyond relaying information of the diagnostic test to generate political support or product buy-in. Yet, community engagement in global health is often messy and not necessarily democratizing [8]. It serves a variety of

purposes, some of which are purely instrumental in ensuring buy-in of stakeholders, others geared at a broader change in unequal power dynamics and politics. The risks and benefits of community engagement activities are often unpredictable and both, empowering and instrumentalising characteristics of community engagement can co-exist within one project [8]. It is therefore essential to analyse and reflect about the process of engagement activities and their intended as well as unintended consequences throughout decision-making processes. The workshops in Kenya and Swaziland aimed to facilitate stakeholder engagement with the introduction of Xpert MTB/RIF Ultra. Positioning qualitative methodology within these workshops offered a unique opportunity to identify perspectives and everyday experiences of key stakeholders and implementers and to contextualize the introduction of a diagnostic test and the interpretation of a particular trade-off within the realities of these everyday experiences. By understanding the specific perspectives and experiences of stakeholders, policy makers will have a better picture of how to improve the contexts within which new diagnostic tests are supposed to function, aiding decision-making, improving implementation as well as patient outcomes. Further research is needed on the intended and unintended consequences of such engagement activities, how findings are being incorporated by decision-makers, and the impact on programmatic implementation.

## Supporting information

**S1 Appendix. Focus group discussion questions.**
(PDF)

**S2 Appendix. Participant responses to success and challenges in TB Diagnosis Kenya.**
(PDF)

## Acknowledgments

We are grateful for the time and insights provided by all the study participants.

## Author Contributions

**Conceptualization:** Kekeletso Kao, Nora Engel.

**Data curation:** Muthoni Mwaura.

**Formal analysis:** Muthoni Mwaura, Nora Engel.

**Methodology:** Muthoni Mwaura, Nora Engel.

**Project administration:** Kekeletso Kao, Jesse Wambugu.

**Resources:** Jesse Wambugu, Andre Trollip, Welile Sikhondze, Eunice Omesa, Sindi Dlamini, Nompumelelo Mzizi, Muyalo Dlamini, Busizwe Sibandze, Brian Dlamini, Heidi Albert.

**Writing – original draft:** Muthoni Mwaura, Kekeletso Kao, Nora Engel.

**Writing – review & editing:** Muthoni Mwaura, Kekeletso Kao, Jesse Wambugu, Heidi Albert, Wybo Dondorp, Nora Engel.

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
