## [Decision Letter · Decision Letter 0]

14 Nov 2019

PONE-D-19-27592

Situating trade-offs: Stakeholder perspectives on overtreatment versus missed diagnosis in transition to Xpert MTB/RIF Ultra in Kenya and Swaziland

PLOS ONE

Dear Dr. Engel,

Thank you for submitting your manuscript to PLOS ONE. After careful consideration, we feel that it has merit but does not fully meet PLOS ONE’s publication criteria as it currently stands. Therefore, we invite you to submit a revised version of the manuscript that addresses the points raised during the review process.

We would appreciate receiving your revised manuscript by Dec 29 2019 11:59PM. To enhance the reproducibility of your results, we recommend that if applicable you deposit your laboratory protocols in protocols.io, where a protocol can be assigned its own identifier (DOI) such that it can be cited independently in the future. For instructions see: http://journals.plos.org/plosone/s/submission-guidelines#loc-laboratory-protocols

We look forward to receiving your revised manuscript.

Kind regards,

HASNAIN SEYED EHTESHAM

Academic Editor

PLOS ONE

Journal Requirements:

1. Thank you for including your competing interests statement; "I have read the journal's policy and the authors of this manuscript have the following competing interests: FIND collaborated with Cepheid and Rutgers to develop the Xpert MTB/RIF Ultra cartridge. FIND also led the evaluation studies to get the cartridge endorsed by WHO thus making uptake of the cartridge in countries easier."

Additional Editor Comments (if provided):

Minor Revision

Reviewers' comments:

Reviewer's Responses to Questions

**Comments to the Author**

1. Is the manuscript technically sound, and do the data support the conclusions?

Reviewer #1: Yes

Reviewer #2: Yes

2. Has the statistical analysis been performed appropriately and rigorously? 

Reviewer #1: Yes

Reviewer #2: N/A

3. Have the authors made all data underlying the findings in their manuscript fully available?

Reviewer #1: Yes

Reviewer #2: Yes

4. Is the manuscript presented in an intelligible fashion and written in standard English?

Reviewer #1: Yes

Reviewer #2: Yes

5. Review Comments to the Author

Reviewer #1: The article is well written and is interesting to check on the efficacy of Gene X pert. The only issue is that the article is too elaborate and needs resizing specially the introduction. Should be modified and resubmitted.

Reviewer #2: Reviewer #: This research article examines the perspectives of stakeholders on the trade-off between over-treatment and missed diagnosis captured during decision-making workshops on the transition from use of Xpert MTB/RIF to diagnose tuberculosis to Xpert MTB/RIF Ultra in Kenya and Swaziland.

The aim of the research was to examine the views and norms of multiple TB stakeholders on the trade-off between overtreatment versus under diagnosis of TB, and to understand the role qualitative research can play in engaging in-country stakeholders during the launch and roll-out of new TB diagnostics.

Positioning qualitative methodology within TB workshops provides a unique opportunity to identify perspectives and everyday experiences of key stakeholders and implementers and to contextualize the introduction of a diagnostic test and the interpretation of a particular trade-off within the realities of these everyday experiences.

Line no 59: to be used as the initial diagnostic test for pulmonary TB in adults.

Thus, I suggest acceptance of this article, as this study is helpful for the stakeholders to make a right decision about a launch and roll-out of new TB diagnostics.

6. PLOS authors have the option to publish the peer review history of their article (what does this mean?). If published, this will include your full peer review and any attached files.

Reviewer #1: No

Reviewer #2: No

---

## [Author Response · Author response to Decision Letter 0]

9 Jan 2020

Responses to reviewer comments PONE-D-19-27592

Reviewer #1: The article is well written and is interesting to check on the efficacy of Gene X pert. The only issue is that the article is too elaborate and needs resizing specially the introduction. Should be modified and resubmitted.

Thank you so much for your supportive comments. We trimmed the text and moved the technical details around the particular trade-off from the introduction into a specific section titled: “The trade off between overtreatment and missed diagnosis”, to make the introduction more readable and to the point.

Reviewer #2: Reviewer #: This research article examines the perspectives of stakeholders on the trade-off between over-treatment and missed diagnosis captured during decision-making workshops on the transition from use of Xpert MTB/RIF to diagnose tuberculosis to Xpert MTB/RIF Ultra in Kenya and Swaziland.

The aim of the research was to examine the views and norms of multiple TB stakeholders on the trade-off between overtreatment versus under diagnosis of TB, and to understand the role qualitative research can play in engaging in-country stakeholders during the launch and roll-out of new TB diagnostics.

Positioning qualitative methodology within TB workshops provides a unique opportunity to identify perspectives and everyday experiences of key stakeholders and implementers and to contextualize the introduction of a diagnostic test and the interpretation of a particular trade-off within the realities of these everyday experiences.

Line no 59: to be used as the initial diagnostic test for pulmonary TB in adults.

Thus, I suggest acceptance of this article, as this study is helpful for the stakeholders to make a right decision about a launch and roll-out of new TB diagnostics.

 Thank you so much for these supportive comments and your suggestion for acceptance. We addressed the typo.

---

## [Editor Report · Decision Letter 1]

22 Jan 2020

Situating trade-offs: Stakeholder perspectives on overtreatment versus missed diagnosis in transition to Xpert MTB/RIF Ultra in Kenya and Swaziland

PONE-D-19-27592R1

Dear Dr. Engel,

We are pleased to inform you that your manuscript has been judged scientifically suitable for publication and will be formally accepted for publication once it complies with all outstanding technical requirements.

With kind regards,

HASNAIN SEYED EHTESHAM

Academic Editor

PLOS ONE

Additional Editor Comments (optional):

The manuscript was sent for minor revision and the authors have redrafted the text based on the suggestion made by the reviewers. I recommend this manuscript for publication.
---

## [Editor Report · Acceptance letter]

7 Feb 2020

PONE-D-19-27592R1 

Situating trade-offs: Stakeholder perspectives on overtreatment versus missed diagnosis in transition to Xpert MTB/RIF Ultra in Kenya and Swaziland 

Dear Dr. Engel:

I am pleased to inform you that your manuscript has been deemed suitable for publication in PLOS ONE. Congratulations! Your manuscript is now with our production department. 

With kind regards,

on behalf of

Prof HASNAIN SEYED EHTESHAM 

Academic Editor

PLOS ONE